# Preparation of SrAl$_2$O$_4$: Eu$^{2+}$, Dy$^{3+}$ Powder by Combustion Method and Application in Anticounterfeiting

Peng Gao [1,†], Jigang Wang [1,†] , Jiao Wu [1], Qingqing Xu [1], Lixue Yang [1], Quanxiao Liu [1,\*], Yuansheng Qi [1,\*] and Zhenjun Li [2,3,\*]

[1] Beijing Key Laboratory of Printing and Packaging Materials and Technology, Beijing Institute of Graphic Communication, Beijing 102600, China; gaopeng08252022@163.com (P.G.); jigangwang@bigc.edu.cn (J.W.); wujiao20220912@163.com (J.W.); xqq1590369@163.com (Q.X.); yanglixue@bigc.edu.cn (L.Y.)

[2] National Center for Nanoscience and Technology, CAS Key Laboratory of Nanophotonic Materials and Devices (Preparatory), Beijing 100190, China

[3] The GBA Research Innovation Institute for Nanotechnology, Guangzhou 510700, China

\* Correspondence: drllqx@163.com (Q.L.); yuansheng-qi@bigc.edu.cn (Y.Q.); lizhenjun@nanoctr.cn (Z.L.)

† These authors contributed equally to this work.

**Abstract:** Green emitting long afterglow phosphor SrAl$_2$O$_4$: Eu$^{2+}$, Dy$^{3+}$ was synthesized via the combustion method. The physical phase analysis was carried out by X-ray diffraction, the results show that the introduction of Eu$^{2+}$ into the lattice of the matrix resulted in a broad green emission centered at 508 nm, which is ascribed to the characteristic $4f^65d^1$ to $4f^7$ electronic dipole allowed transition of Eu$^{2+}$ ions. The doping of Eu$^{2+}$ and Dy$^{3+}$ did not change the physical phase of the crystals. Dy$^{3+}$, as a coactivator, does not emit light itself, but can generate holes to form a trap energy level, which acts as an electron trap center to capture some of the electrons generated by the excitation of Eu$^{2+}$. After excitation has ceased, let them gradually to transfer to the ground state for long afterglow luminescence. Then, we investigate the optical characterizations of different samples excited by X-ray. We found that SrAl$_2$O$_4$: Eu$^{2+}$, 0.5 % Dy$^{3+}$ has this higher luminous intensity and afterglow. Its fluorescence lifetime is about 720 ns, and its quantum yield can reach 15.18%. Through search engine marketing (SEM) and energy dispersive X-ray spectroscopy (EDX), it has been proved that the sample has been successfully synthesized and its component content has been confirmed. The Eg value calculated from the diffuse reflectance spectrum is 4.61eV. The prepared SrAl$_2$O$_4$: Eu$^{2+}$, Dy$^{3+}$ luminescent powder is combined with Polydimethylsiloxane substrate for anticounterfeiting application, which provides a novel idea and method for the development of the anticounterfeiting field.

**Keywords:** SrAl$_2$O$_4$: Eu$^{2+}$, Dy$^{3+}$; long afterglow phosphor; fluorescence lifetime; combustion method; X-ray; PDMS

## 1. Introduction

There have long been inorganic phosphors or luminous materials. The light of these phosphors, which have an inorganic host with a large band gap, is caused by localized transitions between the electronic states of the impurity core, also known as the activator. The chosen dopant ion is the primary factor in determining the spectrum properties of phosphors. There are many afterglow phosphors that have been created, and their emission ranges from ultraviolet to infrared. The aluminates, silicates, or sulfides that have been doped with europium and a suitable trivalent lanthanide are the most effective afterglow phosphors that emit in the visible spectrum. The most encountered afterglow phosphor is SrAl$_2$O$_4$: Eu$^{2+}$, Dy$^{3+}$ [1–5]. In 1968, Palilla [6] et al. discovered the long afterglow characteristics of aluminate system for the first time in the process of studying SrAl$_2$O$_3$: Eu$^{2+}$ luminescence. The long afterglow luminescent materials with excellent luminescence performance of aluminate system are mainly MAl$_2$O$_4$: Eu$^{2+}$, RE$^{3+}$(M = Ca,Sr,Ba;

RE = Dy,Nd,Ho,Er,Pr,Tb, etc.) [7,8], its emission peaks are mainly concentrated in the blue-green light band, the known blue-green long afterglow materials with the best performance are $SrAl_2O_4$: $Eu^{2+}$, $Dy^{3+}$ [9]. The long afterglow material $SrAl_2O_4$: $Eu^{2+}$, $Dy^{3+}$ can emit bright and lasting green long afterglow after excitation with ultraviolet light or X-ray. The afterglow emission peak is located at 520 nm [10], the brightness is 20 times that of the sulfide system, and the afterglow time is more than 24 h [11].

Among strontium aluminates with various stoichiometries, $SrAl_2O_4$ stands out especially as an excellent host lattice for making high emission, long afterglow phosphors by doping with rare earth ions [1]. Therefore, there is a lot of interest in developing such materials by different methods, the most used being the combustion method [12]; other methods include the hydrothermal approach [13] and high-temperature solid-state method [14]. The luminescent materials prepared by hydrothermal method have the advantages of good dispersion, homogeneous phase, high crystallinity and purity of luminescent powder particles and minimum environmental pollution. However, this method requires high equipment and strict technical control of temperature and pressure. High temperature solid state method is the most extensive and earliest method to prepare inorganic luminescent materials. The manufacturing process is relatively mature and the crystal structure formed is good, but its synthesis requires high temperature, long reduction time and long cooling time. However, the combustion method used in this paper, although the optical properties of the sample will be reduced after grinding, the combustion method has a relatively low reaction temperature, short preparation time, simple process, and high purity of the product [12]. It is widely used in experiments.

Currently, phosphors have important applications in many fields. When excited by UV light or an electron beam, $Mn^{2+}$ ions, particularly in $ZnGa_2O_4$ spinel, emit a bright green color that can be used in vacuum fluorescent displays (VFDs), field emission displays (FEDs), and thin-film electroluminescent devices [15]. Using $MgAl_2O_4$ spinel ceramics as an example, the presented positron annihilation lifetime spectra can be used on a broad spectrum of functional nanomaterials with significant porosity [16]. There are several uses for transition metal ions as doping materials in chemistry, physics, biology, material science, and other fields. These uses are related to numerous technologies, including laser technology, nonlinear optics, optical information, phosphor materials, catalysis, and others [17]. Fluorescent powder is also widely used in anticounterfeiting materials, it mainly refers to the basic materials needed to make anticounterfeiting labels, anticounterfeiting trademarks, anticounterfeiting bills, anticounterfeiting cards, etc. At present, the fluorescent anticounterfeiting technology mainly uses photoluminescence. For example, if the paint is mixed with phosphor powder and printed on the product, the pattern is invisible or low-recognition state without ultraviolet light. In external light, it glows. Therefore, it is necessary in order to develop multilevel anticounterfeiting technology [4]. We are also working on some new rare-earth-doped luminescent materials. We obtained a rare-earth-doped luminescent patent this year and carried out technological transformation.

Herein, green emitting phosphors $SrAl_2O_4$, $SrAl_2O_4$: $Eu^{2+}$, and $SrAl_2O_4$: $Eu^{2+}$, $Dy^{3+}$ were prepared successfully via combustion method. The crystal structure, luminescent properties, afterglow behavior, fluorescence lifetime, as well as the quantum efficiency were examined. In addition, the optimal reaction temperature and the optimal amount of Dy ion doping were determined by photoluminescence experiments. Surface morphology of samples with different solvent additions was observed by using scanning electron microscopy. In the end, we combined the prepared $SrAl_2O_4$: $Eu^{2+}$, $Dy^{3+}$ luminescent powder with PDMS elastic substrate for anticounterfeiting application study [12–14,18].

## 2. Materials and Equipment

### 2.1. Materials and Synthesis

$SrAl_2O_4$: $Eu^{2+}$, $Dy^{3+}$ is prepared by a simple combustion method. $SrCO_3$(A.R.), $Al_2O_3$(A.R.), $Eu_2O_3$(A.R.), $Dy_2O_3$(A.R.), and urea are used as raw materials, they were purchased from Tianjin Chemical Reagent Factory. First, $SrCO_3$, $Al_2O_3$, $Eu_2O_3$, and $Dy_2O_3$ are

respectively dissolved in nitric acid to obtain transparent and clear solutions of $Sr(NO_3)_2$ (0.5 mmol/mL), $Al(NO_3)_3 \bullet 9H_2O$ (1 mmol/mL), $Eu(NO_3)_3$ (0.1 mmol/mL), and $Dy(NO_3)_3$ (0.1 mmol/mL). Next, take 3.9 mL of $Sr(NO_3)_2$, 4 mL of $Al(NO_3)_3 \bullet 9H_2O$, 0.4 mL of $Eu(NO_3)_3$, 0.1 mL of $Dy_2O_3$, 2.2 g of urea and a proper amount of deionized water into the crucible, and mix them to obtain a clear solution. Fully stir the mixed solution, burn it in muffle furnace at 800 °C for 2–3 min, hear a "bang" and observe the flame, and take out the product after a moment of cooling. The puffy mushroom cloud like milky white product is obtained after combustion reaction. Green $SrAl_2O_4$: $Eu^{2+}$, $Dy^{3+}$ fluorescent powder can be obtained after cooling and grinding.

The $Dy^{3+}$ doping samples were prepared by the same experimental operation. The doping amount of $Dy(NO_3)_3$(0.1 mmol/mL) was 0.1%, 0.5%, 1%, 2%, 4%, 6%, 8% and 10%, respectively, and 8 samples were obtained. Then three groups of morphology control experiments including EA (Ethyl Acetate), PVA (Polyvinyl alcohol), PEG (Polyethylene glycol), were added respectively to change the micro morphology of the luminescent materials.

*2.2. Instruments*

The crystal structures of the synthesized samples were investigated with an X-ray diffractometer [19] (D/max 2200PC X-ray diffractometer manufactured by Rigaku Corporation). Using the F4700 fluorescence spectrometer [20] for PL test, the emission spectrum and excitation spectrum of the luminescent powder can be drawn to study its spectral properties. The microstructures were observed and compared with a scanning electron microscope (Quanta 250 FEG scanning electron microscope from Hitachi, Japan) [21]. The elemental composition and content were measured by scanning electron microscope-energy meter (Phenom Pro X, Holland). The wavelengths of UV lamps (Manufactured by Hangzhou, China.) is 365 nm. With this excitation source, a continuous fluorescence excitation spectrum in the range of 280–460 nm was measured with a fluorescence spectrometer. The ultraviolet diffuse reflectance spectrum was measured with UV-3600 Ultraviolet Visible Near Infrared Spectrophotometer (Shimadzu Corporation, Kyoto, Japan). The fluorescence lifetime and quantum yield of phosphors were measured by transient steady state fluorescence spectrometer (FLS1000, Edinburgh, UK). Visible light optical fiber spectrometer (FX2000, Shanghai, China) to measure the X-ray fluorescence spectra of different samples and the afterglow luminous intensity by X-ray excited.

### 3. Results and Discussion

*3.1. Phase Identification and Surface Morphology Analysis*

The phase composition and purity of $SrAl_2O_4$: $Eu^{2+}$, $Dy^{3+}$ phosphor at different temperatures and $SrAl_2O_4$: $Eu^{2+}$, $Dy^{3+}$ phosphor doped with Dy ions of different concentrations were investigated by X-ray powder diffraction. Figure 1 shows the X-ray diffraction (XRD) pattern based on samples prepared at different temperatures and different Dy ion doping concentrations. The main reflections were observed at $2\theta$ values 19.951°, 22.740°, 28.386°, 29.275°, 29.922°, and 35.113°. It was found that the diffraction peaks of these samples could match well with the standard $SrAl_2O_4$(JCPDS#34–0379) card. Thus, these peaks can be attributed to the X-ray reflections from monoclinic $SrAl_2O_4$ crystallographic planes (001), (120), (−211), (220), (211), and (013), per the information recorded in JCPDS card number 34-0379 [22]. Strontium aluminate showed prominent peaks at these diffraction angles. The long afterglow phosphorescent substance $SrAl_2O_4$: $Eu^{2+}$, $Dy^{3+}$ was identified as the source of the strong peaks by comparing the phosphor's X-ray pattern with a reference card (JCPDS 34-0379). There were no additional impurity peaks between the observed patterns (black cross) and calculated data (red line), and the refinement parameter value was $R_{wp}$ = 7.448% and $\chi^2$ = 4.61, which confirmed that the $SrAl_2O_4$ host was a single phase. The Rietveld refinement XRD patterns of the sample were carried out by the general structure analysis system (GSAS) software [23].

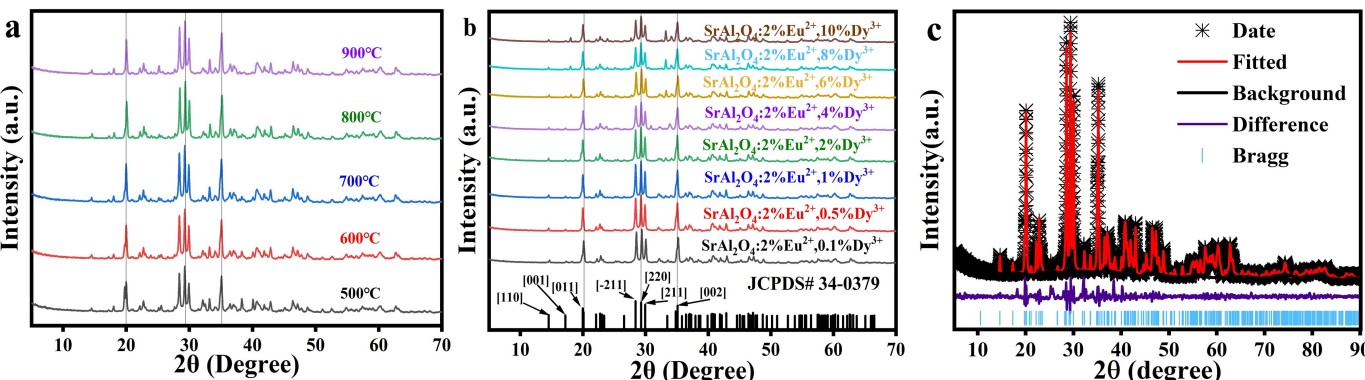

**Figure 1.** (**a**) X-ray diffraction (XRD) investigation based on the samples prepared at different temperatures, (**b**) XRD investigation based on the samples prepared at different Dy ion doping concentration, (**c**) The XRD refinement of $SrAl_2O_4$ host.

High-temperature and low-temperature phases of $SrAl_2O_4$ are composed of hexagonal and monoclinic polymorphs, respectively. It is typically asserted that when $SrAl_2O_4$ is doped with rare-earth ions, only the monoclinic phase exhibits luminescence features [10]. Additionally, the lack of lines from other impurity structures, such as $Sr_4Al_{14}O_{25}$ or $Sr_3Al_2O_6$ low-temperature phases, suggests that the products produced are all $SrAl_2O_4$ monoclinic phase [22]. The ionic radii of the dopants and codopants in $SrAl_2O_4$ determine where they should be positioned. The $Sr^{2+}$ (1.21 Å) ion sites are easily occupied by the $Eu^{2+}$ (1.20 Å) and $Dy^{3+}$ (0.97 Å) ions. $Eu^{3+}$ ions inserted in the $Sr^{2+}$ sites of $SrAl_2O_4$ are easily reduced to $Eu^{2+}$ because $Sr^{2+}$ and $Eu^{2+}$ ions have extremely similar ionic radii [24], $Eu^{2+}$ is expected to replace the $Sr^{2+}$ site in $SrAl_2O_4$. The position and form of the luminescence emission band for Dy ions are not altered by the coactivation of the trivalent rare earth. For $SrAl_2O_4$: $Eu^{2+}$, $Dy^{3+}$, they solely affect the afterglow's duration and intensity, making it much longer and much more intense [25]. Even without $Dy^{3+}$ codoping, the afterglow is there [26]. Because of this, we are confident in stating that $Eu^{2+}$ serves as the luminescence center in these compounds. Thereby demonstrating that $Eu^{2+}$ and $Dy^{3+}$ ions had been successfully doped in $SrAl_2O_4$ matrix.

SEM micrographs of different temperature-controlled synthesis $SrAl_2O_4$: $Eu^{2+}$, $Dy^{3+}$ phosphors (Figure 2) demonstrate this material's granular microstructure. It has a magnification of 1500 times, with an average scale size of about 10 μm. The samples' respective sintering temperatures were 500 °C, 600 °C, 700 °C, 800 °C, and 900 °C. It is evident that as the temperature rises, the phosphor's molecular gap will widen. It can be found that the molecular gap is increased due to the significant amount of gas that the reaction produced after increasing the sintering temperature. Figure 2f shows the elemental composition and content of $SrAl_2O_4$: 2 %$Eu^{2+}$, 0.5 %$Dy^{3+}$ phosphor. The necessary elements of Sr, Al, O, Eu, Dy can be directly recorded in the EDX spectrum. It is clear that the atomic percentages of the $Eu^{2+}$ ion was measured to be about 1.37% and the atomic percentages of the $Dy^{3+}$ ion was measured to be about 0.33%, which is a little lower than the standard value 2%Eu and 0.5%Dy based on the chemical formula due to the influence of oxygen in the air during the tested operations. In the light of the foregoing, it can be suggested that $Eu^{2+}$ and $Dy^{3+}$ activated $SrAl_2O_4$ phosphors have been successfully synthesized in the current study.

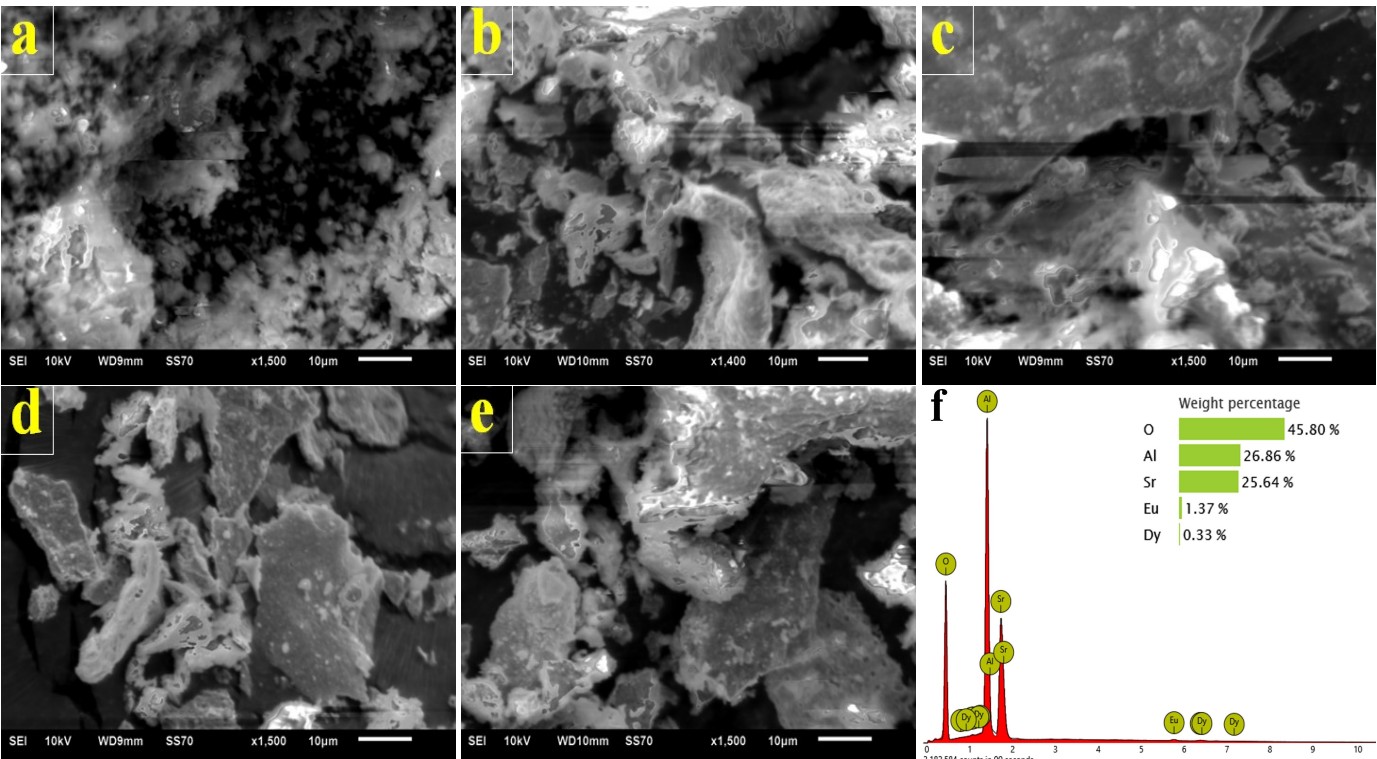

**Figure 2.** Scanning electron microscope (SEM) of the luminescent samples synthesized at (**a**) 500 °C, (**b**) 600 °C, (**c**) 700 °C, (**d**) 800 °C, (**e**) 900 °C, and (**f**) The energy dispersive X-ray spectroscopy (EDX) of the SrAl$_2$O$_4$: Eu$^{2+}$, Dy$^{3+}$.

### 3.2. Optical Characterization

Figure 3a shows the fluorescence emission spectrum of the sample excited by 365 nm UV light when the doping amount of dysprosium ion is 0.5% at the combustion temperature of 800 °C. It can be seen that the emission duration is between 400 nm and 650 nm, and maximum emission peak is at 508 nm. It is basically consistent with the spectroscopic study of SrAl$_2$O$_4$: Eu$^{2+}$, Dy$^{3+}$ single crystal [18]. Figure 3b shows the excitation spectrum after excitation with the emission peak value of 508 nm from the emission spectrum. It can be seen that the peak wavelength is basically at 365 nm, which is consistent with the emission spectrum, indicating that 365 nm is the main excitation wavelength of SrAl$_2$O$_4$:Eu$^{2+}$, Dy$^{3+}$. The stuffed tridymite type structure, in which two distinct Sr$^{2+}$ sites with comparable local distortion are present, is what monoclinic SrAl$_2$O$_4$ belongs to. Because Sr$^{2+}$ and Eu$^{2+}$ have similar ionic radii, Eu$^{2+}$ is expected to occupy Sr$^{2+}$ sites. Eu$^{2+}$ present inside the host lattice is hence responsible for the emission's greenish coloring. The observed excitation of samples is approximately 365 nm which can be due to 4 f$^6$5d$^1$→4 f$^7$ transition of the Eu$^{2+}$ ion. Furthermore, no emission peaks in the red region between the energy levels in the 4f sublayer of Eu$^{3+}$ have been detected, proving that the divalent europium ions of the remaining rare earth ions still exist in the grain boundaries [22]. Dysprosium trivalent ions can form shallow and deep traps in SrAl$_2$O$_4$, while Eu$^{2+}$ ions can only form shallow traps. Dy$^{3+}$ related traps in the range of 0.6 to 1.2 eV from the conduction band are in SrAl$_2$O$_4$:Eu$^{2+}$, Dy$^{3+}$ plays a role in the sensitization of the luminescent center during the luminescence process, which proves that Dy$^{3+}$ has a sensitizing effect on the luminescence of Eu$^{2+}$ [25,27–30].

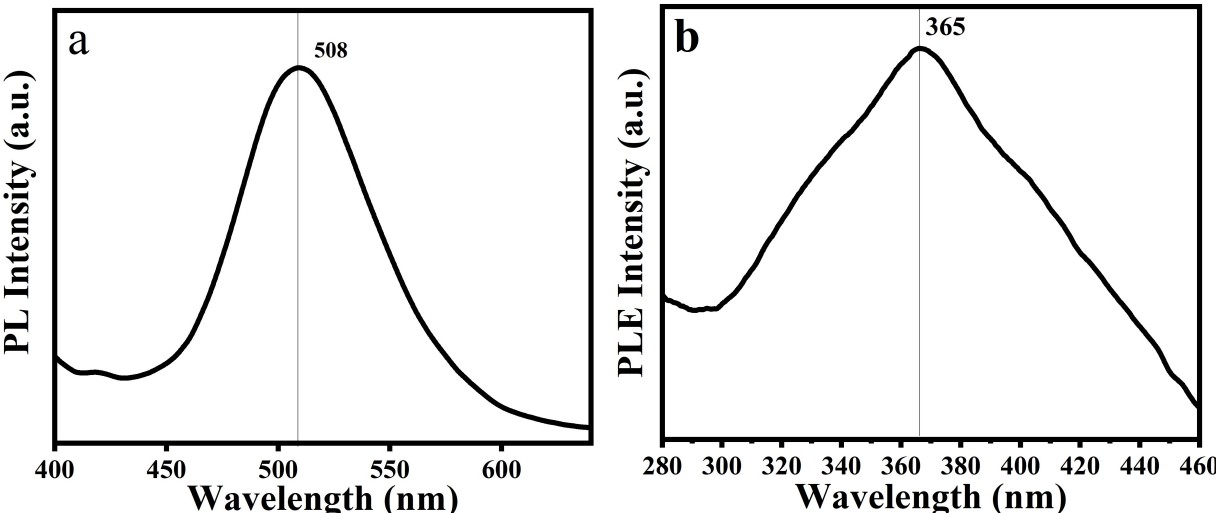

**Figure 3.** (**a**) photoluminescence spectroscopy (PL) emission spectrogram of $SrAl_2O_4$: $2\%Eu^{2+}, 0.5\%Dy^{3+}$ sample was excited at 365 nm, (**b**) Photo Luminescence Excitation (PLE) excitation spectrum of $SrAl_2O_4$: $2\%Eu^{2+}, 0.5\%Dy^{3+}$ sample was detected at 508 nm.

Figure 4a shows the emission spectra of $SrAl_2O_4$: $Eu^{2+}, Dy^{3+}$ samples sintered at different temperatures when excited at 365 nm. The spectral pattern excited at 365 nm is the smoothest and the effect is better. Figure 4b shows the luminescence of the temperature control group under the irradiation of an ultraviolet light at 365 nm. It is evident that the sample have an obvious green light. When the temperature rises from 500 °C to 800 °C, the brightness gradually increases, and the brightness starts to weaken when the temperature increases to 900 °C. It can be found that 800 °C is the optimal reaction temperature for strontium aluminate doped europium and dysprosium, and the brightness decreases when the temperature decreases or increases. It is clear that the combustion temperature during calcination affects the strontium aluminate with europium and dysprosium doping's luminous characteristics.

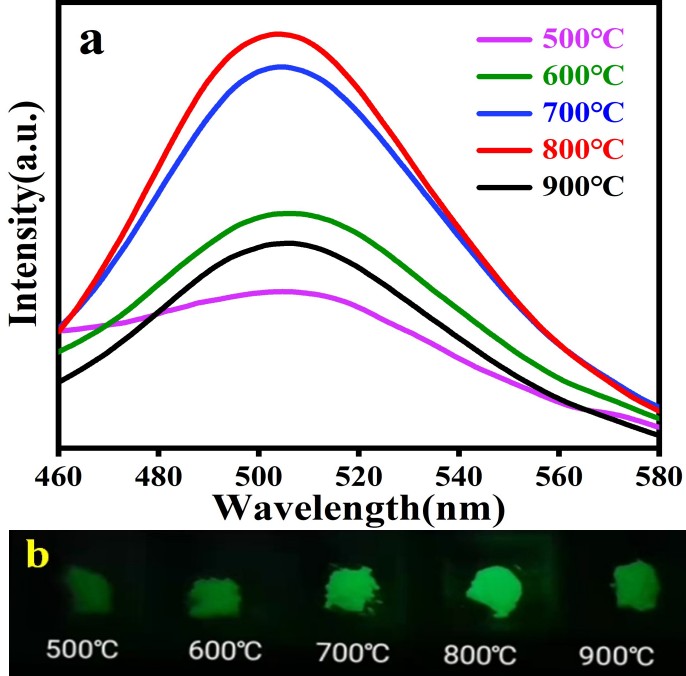

**Figure 4.** (**a**) PL emission spectrum of samples were excited at 365nm; (**b**) Luminescence of temperature control group under 365 nm UV lamp irradiation.

Figure 5a is the emission spectrum of $SrAl_2O_4$: $Eu^{2+}$, $Dy^{3+}$ obtained by changing the doping amount of dysprosium ion at the optimal calcination temperature of 800 °C, and then using the wavelength of 365 nm for excitation. Figure 5b shows the luminescence of 8 groups of luminescent samples doped with different amounts of dysprosium ions under the irradiation of an ultraviolet light at 365 nm. The doping amount of dysprosium ion between the upper left and lower right corners are 0.1%, 0.5%, 1%, 2%, 4%, 6%, 8%, 10%, respectively. It can be seen from Figure 5a,b the luminous intensity is better when the doping content of dysprosium ions sample is 0.5%.

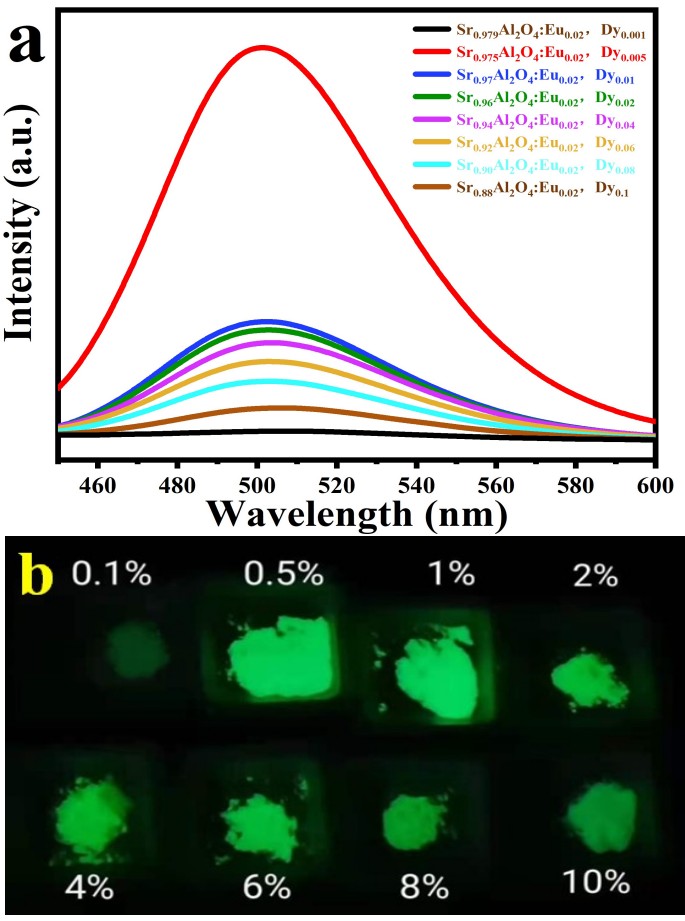

**Figure 5.** (**a**) emission spectra of different $Dy^{3+}$ doping amounts ($\lambda_{em}$ = 365 nm); (**b**) Luminescence of different $Dy^{3+}$ doping amounts under 365 nm UV lamp irradiation.

The production of afterglow characteristics and their strength are intimately related to the traps that the material creates, based on the long afterglow luminescence mechanism. As a coactivator, $Dy^{3+}$ does not emit light on its own, but it can produce holes to create an energy level that serves as an electron trap center, catching part of the electrons emitted by the excitation of $Eu^{2+}$. After the excitation is finished, the trapped electrons are then released by thermal perturbation at the proper rate, enabling them to move into the excited state and then return to the ground state for long afterglow luminescence.When the $Dy^{3+}$ concentration rises, there are more electron traps available, which allows for a large amount of electrons to be released gradually after the excitation has ended, increasing the brightness of the afterglow. The electron trap density, on the other hand, becomes excessively high and the corresponding distance between the traps shortens when the concentration of $Dy^{3+}$ exceeds a particular amount. The phenomena of afterglow concentration quenching will then take place as the carriers move between the traps. The brightness of the afterglow finally starts to fade. The findings from experiments indicate that a $Dy^{3+}$ concentration of 0.5% results in the optimum afterglow performance.

Figure 6 shows the X-ray fluorescence spectra of different samples. It can be seen that the luminous intensity of strontium aluminate is weak, and its luminescence can hardly be found, as shown in Figure 7a. When Eu ion is added, it can be found with a slight green light, as shown in Figure 7b. When Eu and Dy ions are added simultaneously, compared to Figure 7a, green light can be clearly seen. When 0.5% Dy ion is added, its luminous intensity is the highest, as shown in Figure 7d. Its luminous intensity is about 13.3 times of $SrAl_2O_4$ and 3.05 times of $SrAl_2O_4$: $Eu^{2+}$. Therefore, when Eu ions and 0.5%Dy ions are doped, the X Ray luminescence performance is better. The photoluminescence under X-ray excitation is basically consistent with that under ultraviolet irradiation.

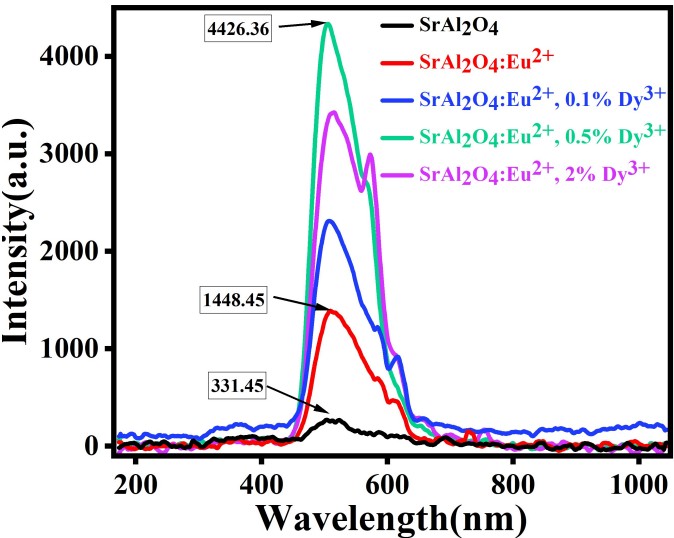

**Figure 6.** X-ray fluorescence spectra of different samples.

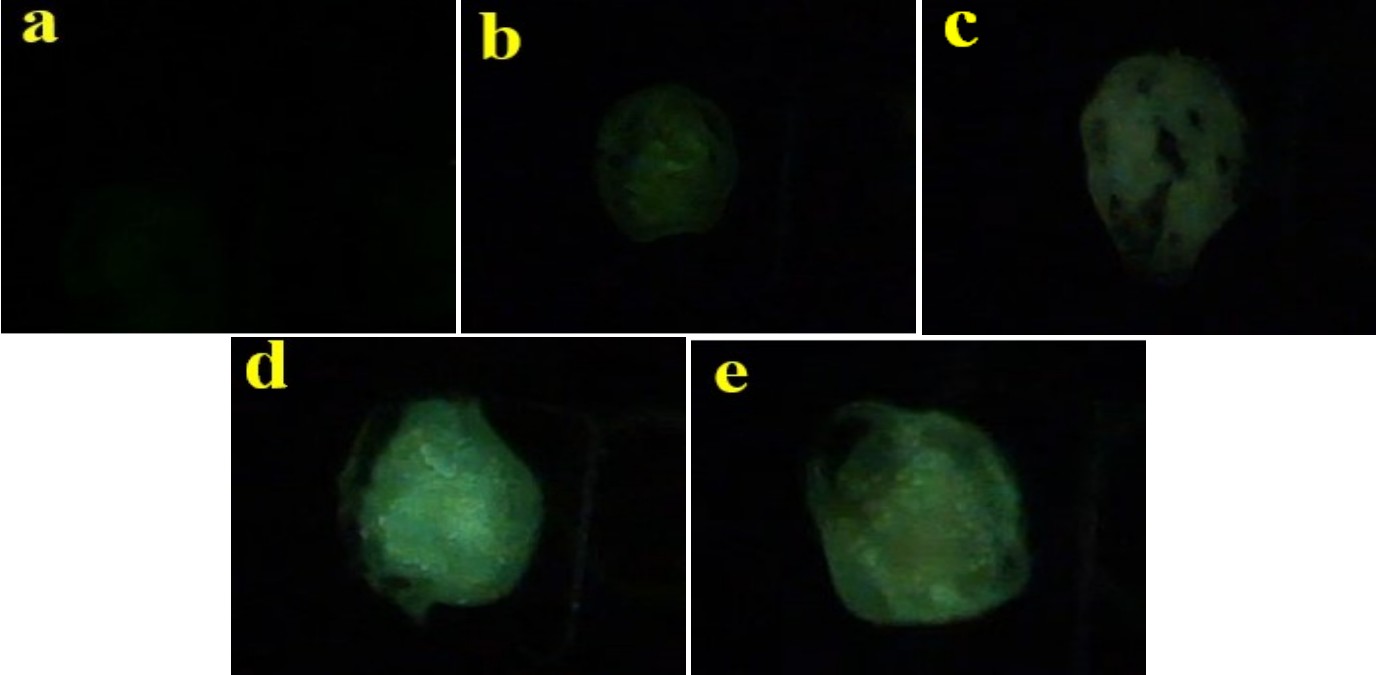

**Figure 7.** Optical imaging pictures of 5 samples under X-ray excitation (**a**) $SrAl_2O_4$, (**b**) $SrAl_2O_4$: $Eu^{2+}$, (**c**) $SrAl_2O_4$: $Eu^{2+}$, 0.1 %$Dy^{3+}$, (**d**) $SrAl_2O_4$: $Eu^{2+}$, 0.5 %$Dy^{3+}$, and (**e**) $SrAl_2O_4$: $Eu^{2+}$, 2%$Dy^{3+}$.

Figure 8 shows the afterglow attenuation curves of different samples under X-ray irradiation. It can be seen that afterglow luminous intensity of $SrAl_2O_4$ and $SrAl_2O_4$:

Eu$^{2+}$ is low, the most interesting phenomenon is that after doping Dy ions, the afterglow luminous intensity has been significantly improved. The samples' luminescence decay process is nonexponential. Generally, in aluminate phosphors, the luminescent centers are Eu$^{2+}$ ions and the traps are Dy$^{3+}$ ions [31]. The trap energy level produced by doping Eu$^{2+}$ and Dy$^{3+}$ ions in crystals usually causes the long afterglow property [13]. The quick decay is caused by the electron's short survival time in Eu$^{2+}$ and Eu$^{2+}$ions immediately transition from $4f^65d^1 \rightarrow 4f^7$ level. The presence of Dy$^{3+}$ ions in the host lattice and the deep trap energy center of Dy$^{3+}$ are attributed to the slower decay process, which is only significant in SrAl$_2$O$_4$: Eu$^{2+}$, Dy$^{3+}$ samples [32].

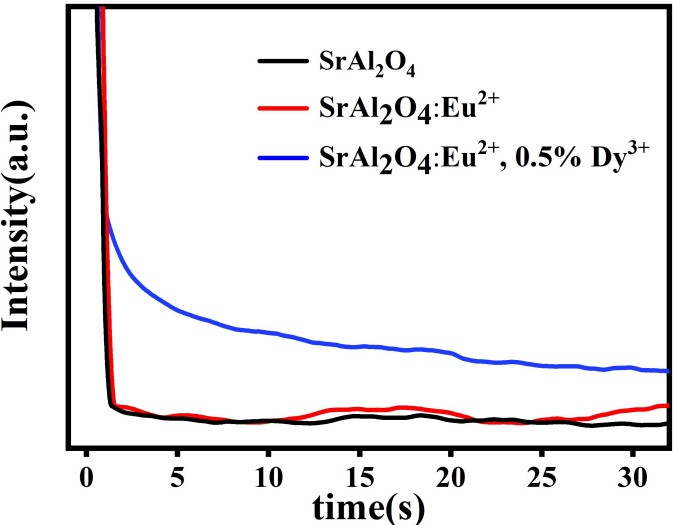

**Figure 8.** Afterglow of each sample under X-ray.

The long afterglow mechanism could be a hole trapped-transported-detrapped process [30]. The Dy$^{3+}$ acts as a trap of holes (trap levels), with the trap levels located between the excited and ground states of Eu$^{2+}$. Following UV or X-ray excitation, electron and hole pairs are produced, and some free holes transported in the conduction band are captured by the Dy$^{3+}$ traps. When the excitation source was removed, some holes trapped by the Dy$^{3+}$ traps were thermally released slowly and relaxed to the excited state of Eu$^{2+}$, before returning to the ground state of Eu$^{2+}$ and emitting light. This is why phosphor sustained a long afterglow duration at a poor luminescent level of intensity [33]. Herein, an explanation for the afterglow of these SrAl$_2$O$_4$: Eu$^{2+}$, Dy$^{3+}$ phosphors was put out (Figure 9).

Fluorescence lifetime refers to the time elapsed when the fluorescence intensity decreases to 1/e of the fluorescence intensity at the time of stopping excitation when the excitation source is turned off. The fluorescence lifetime curve of SrAl$_2$O$_4$ and SrAl$_2$O$_4$: 0.02 Eu$^{2+}$, xDy$^{3+}$(x = 0, 0.001, 0.01, 0.04, 0.06, 0.08, and 0.1) samples was measured under the xenon lamp and the curve was well fitted, as shown in Figure 10. The fluorescence lifetime curves are divided into two regimes: the initial rapid decay and the subsequent slow decay. In general, the following exponential relationships can best fit the fluorescence lifetime decay curve [33]:

$$I = I_0 + A_1 e^{-\frac{t}{\tau_1}} + A_2 e^{-\frac{t}{\tau_2}} \tag{1}$$

where I is the phosphorescence intensity, $A_1$ and $A_2$ are the constants, t is the time, $\tau_1$, and $\tau_2$ are the decay times for the exponential components. Using the software of Origin 2022, the values of $\tau_1$ and $\tau_2$ of the phosphors can be obtained, as listed in Table 1. The average lifetimes can also be obtained by the Formula (2) [34]. The detail parameters are listed in Table 1.

$$\tau_{avg} = \frac{A_1 \tau_1^2 + A_2 \tau_2^2}{A_1 \tau_1 + A_2 \tau_2} \tag{2}$$

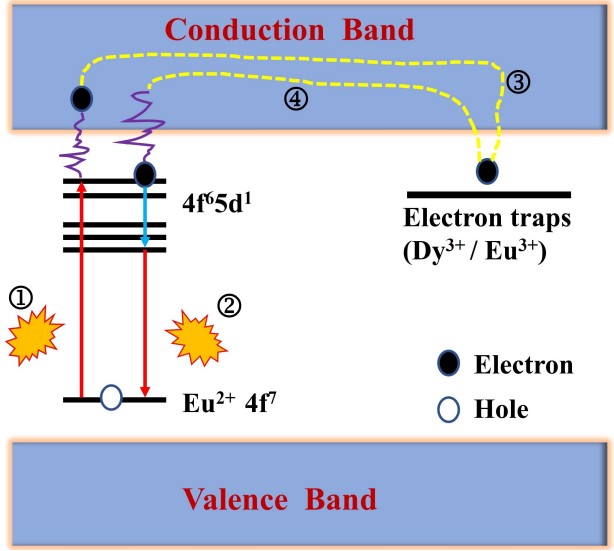

①: The electrons in $4f^7$ energy level of $Eu^{2+}$ion were photoionized to $4f^65d^1$ excited state under light source excitation.

②: The $Eu^{2+}$ ions in the unstable $4f^65d^1$ excited state partially emit photons to return ground state and partially release electrons to the conduction band.

③: The remaining few excited electrons transferred through conduction band would be caught by the trap center.

④: The gradual thermal release of electrons from traps led creation of $Eu^{2+}$ excited state due to electron and $Eu^{3+}$ interaction and this excited $Eu^{2+}$ is the origin of long afterglow observed.

**Figure 9.** Schematic diagram of afterglow mechanism for $SrAl_2O_4$: $Eu^{2+}$, $Dy^{3+}$.

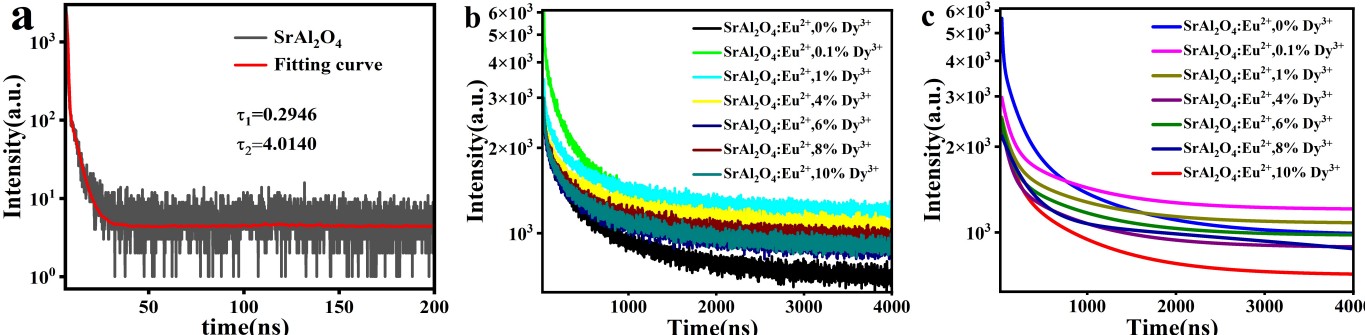

**Figure 10.** (**a**) Fluorescence lifetime decay curve and fitting curve of $SrAl_2O_4$, (**b**) Fluorescence lifetime decay curves of different samples of $SrAl_2O_4$: $0.02\,Eu^{2+}$, $xDy^{3+}$ (x = 0, 0.001, 0.01, 0.04, 0.06, 0.08, and 0.1), (**c**) Fluorescence lifetime decay fitting curves for different samples of $SrAl_2O_4$: $0.02\,Eu^{2+}$, $xDy^{3+}$ (x = 0, 0.001, 0.01, 0.04, 0.06, 0.08, and 0.1).

**Table 1.** Parameters of fluorescence lifetime decay curves for different samples.

| Sample | Decay Lifetimes (ns) | | | | |
|---|---|---|---|---|---|
| | $A_1$ | $A_2$ | $\tau_1$ | $\tau_2$ | $\tau_{avg}$ |
| $SrAl_2O_4$ | 0.088 | 0.002 | 0.2946 | 4.0140 | 1.174 |
| $SrAl_2O_4$: $Eu^{2+}$, 0 %$Dy^{3+}$ | 822.640 | 747.737 | 117.708 | 812.838 | 717.311 |
| $SrAl_2O_4$: $Eu^{2+}$, 0.1 %$Dy^{3+}$ | 1924.719 | 1110.173 | 180.771 | 907.019 | 720.518 |
| $SrAl_2O_4$: $Eu^{2+}$, 1 %$Dy^{3+}$ | 1021.863 | 753.378 | 97.536 | 813.502 | 713.355 |
| $SrAl_2O_4$: $Eu^{2+}$, 4 %$Dy^{3+}$ | 820.139 | 640.820 | 106.750 | 828.777 | 726.596 |
| $SrAl_2O_4$: $Eu^{2+}$, 6 %$Dy^{3+}$ | 890.813 | 628.545 | 88.892 | 798.896 | 702.182 |
| $SrAl_2O_4$: $Eu^{2+}$, 8 %$Dy^{3+}$ | 877.156 | 633.436 | 94.751 | 818.088 | 718.112 |
| $SrAl_2O_4$: $Eu^{2+}$, 10 %$Dy^{3+}$ | 1147.305 | 648.023 | 110.980 | 852.331 | 713.445 |

Through the Table 1, we obtain the fluorescence lifetimes of the series of $SrAl_2O_4$ and $SrAl_2O_4$: $0.02\,Eu^{2+}, xDy^{3+}$ (x = 0, 0.001, 0.01, 0.04, 0.06, 0.08, and 0.1) samples as 1.174, 717.311, 720.518, 713.355, 726.596, 702.182, 718.112, 713.445 ns, respectively. When Eu and Dy are not added to the $SrAl_2O_4$, it can be seen that its fluorescence lifetime is almost zero. When $Eu^{2+}$ ions is doped into $SrAl_2O_4$, the intuitive result is that the fluorescence lifetime is increased. When $Dy^{3+}$ ions is doped into $SrAl_2O_4$: $Eu^{2+}$, has little impact on its fluorescence lifetime. This may be due to that dysprosium addition will cause lattice distortion, which may affect the luminescence of europium and reduce the lifetime of europium. However, this process may be accompanied by energy transfer from Dy to Eu, which in turn can improve the fluorescence lifetime of Eu. There is already a large amount of literature [2,35,36] to prove that if the energy transfer occurs in a system, the fluorescent decay lifetimes of sensitizer should decline with the increase of activator content. But in this experiment, sensitizer is Dy and activator is Eu. We studied the relationship between the fluorescence decay lifetime of sensitizer and activator. However, we found that the addition of Dy ions does not affect their fluorescence lifetime.

To investigate the absorption properties and band gap value of the obtained $SrAl_2O_4$: $Eu^{2+}, Dy^{3+}$, UV–vis diffuse reflection spectra is conducted and the spectrum. The UV–vis diffuse reflectance spectra of synthetic $SrAl_2O_4$: $Eu^{2+}, Dy^{3+}$ recorded in the 200–800 nm region is shown in Figure 11. The band gap could be calculated based on UV–vis diffuse reflectance result using Tauc's equation that given by [37]:

$$(\alpha h v)^2 = A(h v - E_g) \tag{3}$$

where $hv$ is the photon energy, $A$ is a material-dependent constant and $E_g$ is the energy band gap. $\alpha$ is the optical absorption coefficient, it can be obtained by the Kubelka–Munk function [37]:

$$\alpha = \frac{1 - R^2}{2R} \tag{4}$$

$R$ is the observed reflectance in the diffuse reflectance spectrum. Inset of Figure 11 displays the plot of $(\alpha hv)^2$ against (Energy), which enables extrapolation of the straight-line graph at $(\alpha hv)^2 = 0$ to compute the energy gap. The synthesized $SrAl_2O_4$: $Eu^{2+}, Dy^{3+}$'s band gap was calculated to be 4.61 eV, which is reasonable and in reasonable accord with the previously published literature results [11,37,38].

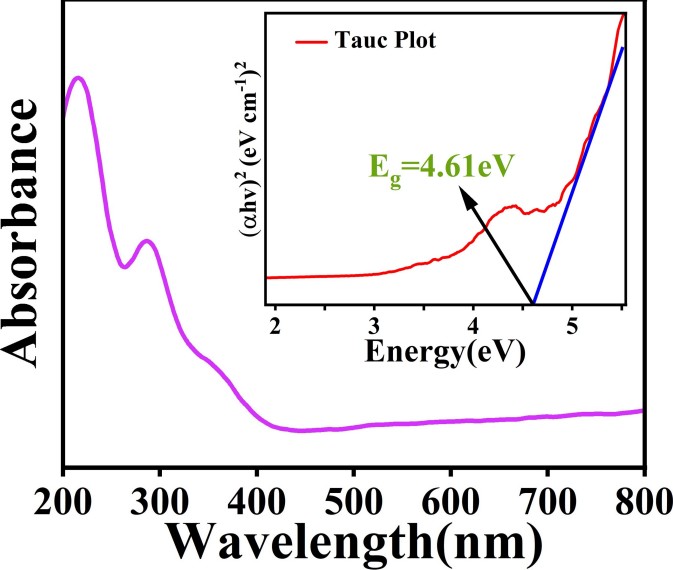

**Figure 11.** UV absorption diffuse reflection spectrum of the prepared $SrAl_2O_4$: $Eu^{2+}, Dy^{3+}$ (inset:the band gap values).

*3.3. Luminescence and Morphology of SrAl$_2$O$_4$: Eu$^{2+}$, Dy$^{3+}$ in Different Solvents*

Figure 12a shows the emission spectra of SrAl$_2$O$_4$: Eu$^{2+}$, Dy$^{3+}$ luminescent materials with three solvents when excited by ultraviolet light with a wavelength of 365nm. The positions of three broadband peaks are basically the same, and the peak with Ethyl Acetate(EA) solvent is the highest, followed by the sample with Polyethylene glycol(PEG) solvent, and the lowest is the sample with Polyvinyl alcohol(PVA) solvent. Figure 12b shows the XRD diffraction patterns of SrAl$_2$O$_4$: Eu$^{2+}$, Dy$^{3+}$ samples with three solvents added and SrAl$_2$O$_4$: Eu$^{2+}$, Dy$^{3+}$ samples without solvents added. From top to bottom, the XRD diffraction patterns of the samples with EA, PVA, PEG solvent and without solvent are shown respectively. Through waveform comparison, it can be found that the three samples have three main crystal diffraction peaks, and the intensity and position are almost the same as those without solvents, and the crystal structure is consistent, indicating that the addition of solvent does not change its crystal structure.

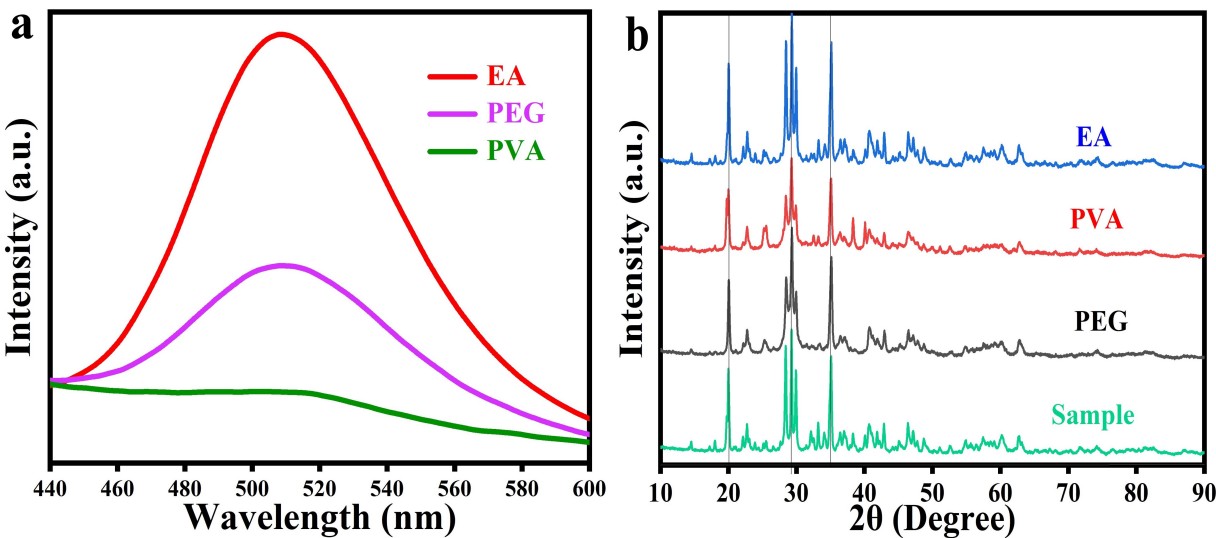

**Figure 12.** (**a**) Emission spectra of phosphors with different solvents ($\lambda_{em}$ = 365 nm), (**b**) XRD patterns of samples with different solvents.

SEM micrographs of different solvents synthesis SrAl$_2$O$_4$: Eu$^{2+}$, Dy$^{3+}$ phosphors (Figure 13) demonstrate this material's granular microstructure. It has a magnification of 1500 times, with an average scale size of about 10 μm. Solvent has the function of micellization, which can change its surface tension. From the comparison of the three images, it can be observed that their morphology has changed. It can be observed that the gap from EA to PVA and PEG samples has become smaller gradually, and the solvent has changed the dispersion degree between particles. Compared with the SEM image of the sample prepared without adding solvent in Figure 2, different solvent will also change the gap of the sample, which is accompanied by the shape.

*3.4. Quantum Yield*

The above samples were measured at the exceptional range from 350.00 nm to 380.00 nm and luminescence range from 380.00 nm to 800.00 nm. It can be seen that the quantum yield of strontium aluminate is particularly low without doping any rare earth ions, as shown in Figure 14. When Eu ions are doped, the quantum yield of strontium aluminate can be improved. When Eu ions and Dy ions are both doped, the quantum yield of strontium aluminate will need to be further improved. After doping rare earth elements, strontium aluminate has a higher quantum yield, indicating that it is a appropriate matrix for rare earth ion doping. Generally, we use the following formula to calculate the quantum yield [39]:

$$PLQY = \frac{N_{emission}}{N_{absorption}} = \frac{\int \frac{\lambda}{hc}\{I_{em}^{pro}(\lambda) - I_{em}^{ref}(\lambda)\}d\lambda}{\int \frac{\lambda}{hc}\{I_{ex}^{ref}(\lambda) - I_{ex}^{pro}(\lambda)\}d\lambda} \tag{5}$$

The $N_{absorption}$ is the number of photons absorbed by the sample, $N_{emission}$ is the number of photons emitted from the sample, $\lambda$ is the wavelength, $h$ is Planck's constant, $c$ is the velocity of light, $I_{ex}^{pro}$ and $I_{ex}^{ref}$ are the integrated intensities of the excitation light with the sample and reference sample, respectively, $I_{em}^{pro}$ and $I_{em}^{ref}$ are the mission intensities with the sample and reference sample, respectively.

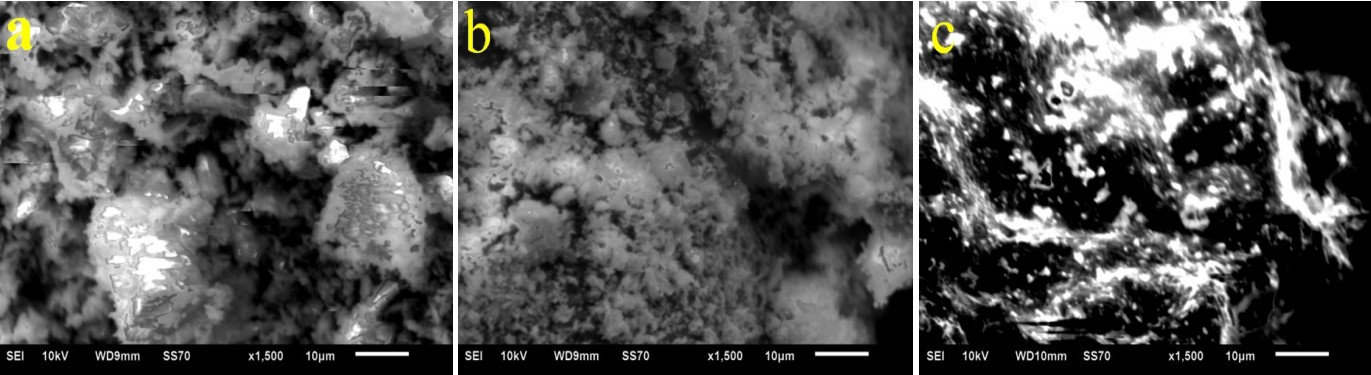

**Figure 13.** SEM images of the luminescent samples synthesized after adding (**a**) EA; (**b**) PVA; (**c**) PEG.

In this paper, red is a white board, which is used to compare with the sample wire to obtain the quantum yield. Black represents the sample line. The quantum efficiency is the area difference between the emission section (450 nm–650 nm) and the excitation section (350 nm–380 nm). These figures are logarithmic to the ordinate, so the excitation section is small. If the logarithm is not taken, the intensity of the white board at the excitation end will be several orders of magnitude higher than that of the sample. Herein, Figure 14d has the highest quantum yield. However, compared with other substances [40], its quantum yield is relatively low, this is also where efforts should be made in the future. How to further improve the quantum yield of strontium aluminate significantly needs more research.

*3.5. Preparation and Morphology of Pdms Composites*

Polydimethylsiloxane (PDMS) is well known as organosilicon which is a kind of polymer organosilicon compound material. It is optically transparent, nontoxic, excellent water resistance, oxidation resistance, chemical inertia, and thermal stability. The curing agent, also known as a hardener, is a substance that can control and enhance the curing reaction.

To further study the anticounterfeiting properties of phosphors, we combined rare-earth doped $SrAl_2O_4$: $Eu^{2+}$, $Dy^{3+}$ luminescent powders with PDMS elastic substrates. $SrAl_2O_4$: $Eu^{2+}$, $Dy^{3+}$ luminescent powder materials were prepared by the combustion method. The silicone rubber selected for use is Dow Corning 184, which contains PDMS based resin and curing agent. They are mixed in the ratio of 10:1, one part is reserved for use, and the other part is added with $SrAl_2O_4$: $Eu^{2+}$, $Dy^{3+}$ powder. Stir evenly to make them fully fuse into milky white viscous emulsion. Then pour it into the mold with "mL" letter. Because the mixing will mix air in the liquid, which will affect the anticounterfeiting effect, it needs to be placed in the vacuum drying oven for vacuum drying. The vacuum degree of the vacuum drying oven is set to 0.1. It can be observed that when vacuuming, a large number of bubbles in the emulsion will be slowly pumped to the surface and then dissolve and disappear. This process generally requires 20min to get a transparent mixture without bubbles. Then put it into a muffle furnace at 120 °C for heating, solidify it into shape, take it out after 40min and cool it in cold water. This process is shown in Figure 15a. Next, PDMS elastic substrate shall be made. After cooling, the formed solid mixture material is

buckled from the letter mold and placed in a rectangular mold. The PDMS prepared for use is poured into the rectangular mold. Repeating the previous operation. First, remove the bubbles in the milk with a vacuum drying oven, and then heat it in a muffle furnace. Finally, the PDMS composite material sample is obtained.

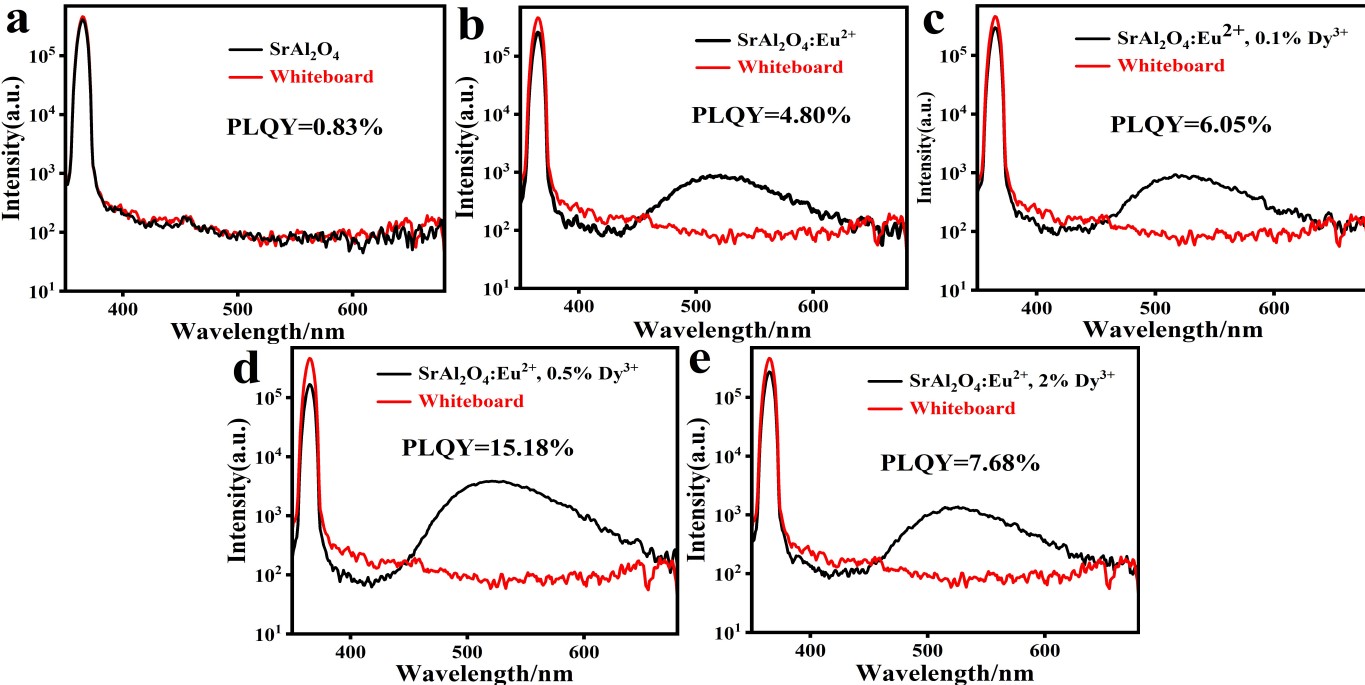

**Figure 14.** Fluorescence spectra of different samples and quantum yield (**a**) $SrAl_2O_4$, (**b**) $SrAl_2O_4$: $Eu^{2+}$ , (**c**) $SrAl_2O_4$: $Eu^{2+}$, 0.1 %$Dy^{3+}$, (**d**) $SrAl_2O_4$: $Eu^{2+}$, 0.5 %$Dy^{3+}$, and (**e**) $SrAl_2O_4$: $Eu^{2+}$, 2%$Dy^{3+}$.

The luminescent powder is wrapped by PDMS. It has the characteristics of flexible materials which can be stretched and can also be returned to its original shape. The composite material is almost transparent in the sunlight (Figure 15b), while when illuminated by the ultraviolet lamp, the "ML" letter added with $SrAl_2O_4$: $Eu^{2+}$, $Dy^{3+}$ luminous powder will appear green (Figure 15c). Moreover, PDMS is low cost and simple used, so we can easily use it in anticounterfeiting packaging.

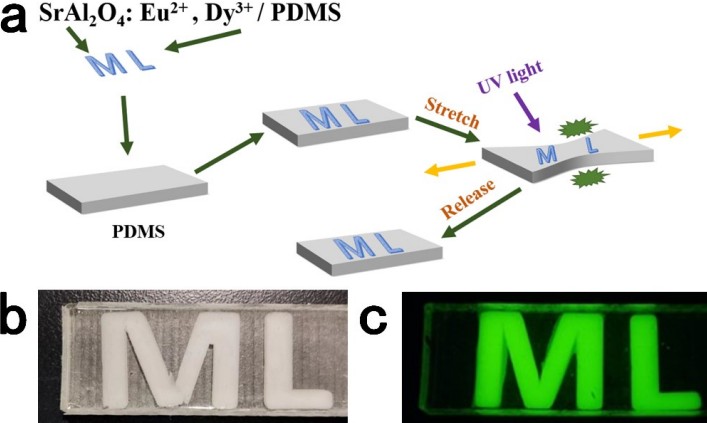

**Figure 15.** (**a**) Preparation process of PDMS composites and its elastic properties, (**b**) PDMS composites in natural light, (**c**) PDMS composite under UV lamp irradiation.

Figure 16a,c are SEM images without fluorescent powder, and Figure 16b,d are SEM images with fluorescent powder. The sintering temperature of the samples was 120 °C. When the magnification reaches 200 μm, by observing the surface of PDMS composite

that small particles can be barely seen on the surface doped with fluorescent powder. Thereupon, cross section is cut by brittle fracture of the liquid nitrogen method. It can be clearly observed in the Figure 16c that the cross section without fluorescent powder is smooth. It can be observed from Figure 16d that small particles and bubbles appear on the cross section of fluorescent powder. This result shows that the fluorescent powder can be applied in the anticounterfeiting of PDMS.

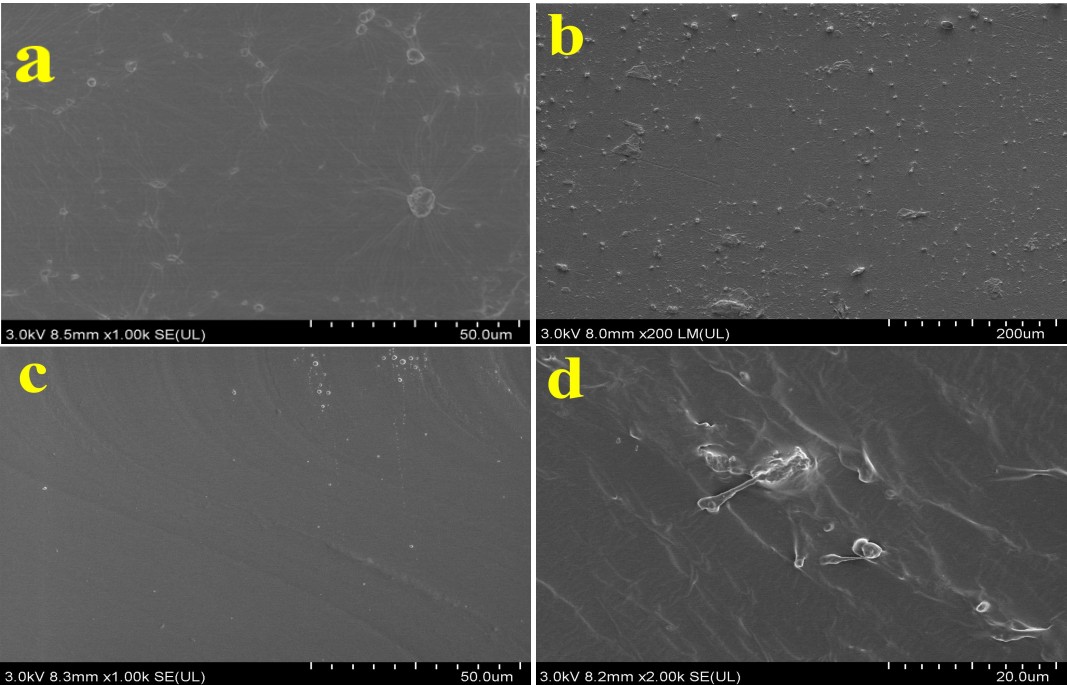

**Figure 16.** (**a**) SEM surface morphology of PDMS composite without fluorescent powder and the average size of about 50 μm, (**b**) SEM surface morphology of PDMS composite with fluorescent powder and the average size of about 200 μm (**c**) SEM sectional morphology of PDMS composite without fluorescent powder and the average size of about 50 μm, (**d**) SEM sectional morphology of PDMS composite with fluorescent powder and the average size of about 20 μm.

### 4. Conclusions

In summary, $SrAl_2O_4$: $Eu^{2+}, Dy^{3+}$, a phosphor with long afterglow property, was successfully prepared by the combustion method. XRD analysis showed that $Eu^{2+}$ and $Dy^{3+}$ were successfully doped into the $SrAl_2O_4$ lattice. Under the excitation of X-rays and ultraviolet rays, $SrAl_2O_4$: $Eu^{2+}, Dy^{3+}$ phosphors exhibit green luminescence, which was ascribed to the $4f^65d^1$ to $4f^7$ transition of $Eu^{2+}$. $Eu^{2+}$ ion is the luminescent centers and $Dy^{3+}$ has a sensitizing effect on the luminescence of $Eu^{2+}$. Changes in the content of Dy ions have little impact on the fluorescence lifetime of the sample, which is approximately 720 ns. The optimum doping concentrations of $Dy^{3+}$ is 0.5%, and its quantum yield is 15.18%. SEM test shows that adding different surfactants EA, PVA, and PEG can change the morphology of luminescent materials, but does not change its crystal structure. EDX experiments have confirmed that the sample has been successfully synthesized and its component content has been confirmed. The Eg of the sample was calculated to be 4.61eV by UV diffuse reflectance spectroscopy. The prepared $SrAl_2O_4$: $Eu^{2+}, Dy^{3+}$ luminescent powder is successfully combined with PDMS means it can be well applied for anticounterfeiting application.

**Author Contributions:** P.G., J.W. (Jigang Wang), J.W. (Jiao Wu) and Q.X.; designed and carried out the experiment. P.G., J.W. (Jigang Wang), L.Y. and Q.L.; processed the test data. With the help of Y.Q. and Z.L.; wrote and completed the paper. All authors have read and agreed to the published version of the manuscript.

**Funding:** This research is supported by Beijing Natural Science Foundation(No. 2202018), General Project of Beijing Municipal Education Commission Science and Technology Program (No. KM202010015004), Research and development of intelligent packaging for cultural relics (Ed202001), Construction and application transformation of cross media cloud platform for printing and packaging anticounterfeiting and traceability (27170121005), National Natural Science Foundation of China (No. 21604005, Grant No. 52072084), the general project of fundamental research of BIGC (No. Ed202208, Eb202001, 20190122041, 22150122042, 22150122034, 27170122012), Initial funding for the Doctoral Program of BIGC (No. 27170121001/002), and the general project of science and technology of Beijing Municipal Education Commission (No. KM202110015008), the Key Area Research and Development Program of Guangdong Province (Grant no. 2020B0101020002), the GBA National Institute for Nanotechnology Innovation (Grant No. 2020GN0106), the National Key R&D Program of China (Grant No. 2021YFC2802000), National key research and development program(2019YFB1707202).

**Institutional Review Board Statement:** Not applicable.

**Informed Consent Statement:** Not applicable.

**Data Availability Statement:** Research data are not shared.

**Conflicts of Interest:** The authors declare no conflict of interest.

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
