# Peer review of "Preparation of SrAl2O4: Eu2+, Dy3+ Powder by Combustion Method and Application in Anticounterfeiting"

_coatings, doi:10.3390/coatings13040808_

Round 1
Reviewer 1 Report
The manuscript "Preparation of SrAl2O4: Eu2+, Dy3+ Powder by combustion method and application in anti-counterfeiting" shows some interesting results especially in the optimization process either temperature of Dy concentration. There are some points that needs to be clarify as well the manuscript is overloaded and need be better structured.
1. The combustion method for same combination SrAl2O4: Eu2+, Dy3+ has been published before not cited in this manuscript (Nguyen Manh Son et al 2009 J. Phys.: Conf. Ser. 187 012017). please add this reference
2. The section 3.5 line 317 - 342 should be shown in the introduction as it makes less sense in the part PDMS was introduced. Also Figure 18 and 19 did not show any relevant results, the reviewer suggest to create supplementary section and include those. Also the authors talk about cross section at Figure 20 but there is no Figure 20. Figure 15-17 can be combined to Figure 15a-c as its the same results and would look much better together.
3. Figure 2 SEM different temperature. It looks like that at higher temperature more bulk appearance can be seen in comparison to low temperature. Has the porosity of the material any effect in the optical properties? Is it possible to define those microstructure either over imagej or BET?
4. The authors said referring to Figure 4a and b that 800 oC was the best while at 900 degree its luminescence is reduced as can be seen clearly at Figure 4b. But why does such happens as no explanation provided in the text. please give some meaningful explanation.
5. Figure 7. The authors says that luminescence is improved but that can barely been seen in Figure 7b. Please enhance image or reformulate the sentence.
Some minor mistakes and text editing
Page 5 line 122. please add a reference. line 127 "ionsthe" please correct it
Page 8 line 168 "4f to 4f" what does that means hence its a bit confusing
Page 18 line 300. the equation 4 look different from others given in this script. It looks copied. please use right format
Reviewer 2 Report
This is a rather interesting and good paper that probably can be recommended for publication, but clarifying and detailing some parts of the text.
1. It is not clear why this article was submitted to Coatings, because this word itself does not even occur in the text of the article.
2. It is important to note that these and similar materials have many important applications as optical materials, or materials for nuclear/fusion technology, such as diagnostic/detector components and materials for shielding etc.
See for example:
Luchechko, A., Zhydachevskyy, Y., Ubizskii, S. et al. Afterglow, TL and OSL properties of Mn2+-doped ZnGa2O4 phosphor. Sci Rep 9, 9544 (2019). https://doi.org/10.1038/s41598-019-45869-7
Klym, H.; Karbovnyk, I.; Piskunov, S.; Popov, A.I. Positron Annihilation Lifetime Spectroscopy Insight on Free Volume Conversion of Nanostructured MgAl2O4 Ceramics. Nanomaterials 2021, 11, 3373. https://doi.org/10.3390/nano11123373
Etimie, EL Andreici, Nicolae M. Avram, and Mikhail G. Brik. "The dd transitions and ligand field parameters for Cr3+/Co2+ doped (Mg, Zn)Al2O4: Multi-reference Ab initio investigations." Optical Materials: X 16 (2022): 100188. https://doi.org/10.1016/j.omx.2022.100188
3. For optical materials, Eg band gap data are absolutely important, but this information is missing for considered phosphor.
4. Table 1. Absolutely limitless accuracy, which in principle is unattainable. What does 0.0000001 nm mean ??? Note, that the dimensions of the hydrogen atom are close to 0.1 nm !!!
5. Line 149. What does “molecular gap” mean?
6. Fig.2. How do these pictures behave over time?
7. Fig. 3. “…was excited at 508nm”. Here, must be “was detected at 508nm”.
8. Fig. 6. It would be useful to compare the shape of these spectra by normalizing them . Moreover, it is useful to decompose them into Gaussian components so that all trends can be seen.
9. It is known that the MgAl2O4 spinel can be of different stoichiometry. How was the stoichiometry checked in this work SrAl2O4?
1. In the conclusions, it is necessary to formulate more clearly what new data on the studied materials were obtained in this work?
In general, the manuscript is interesting and can be recommended for publication after constructive reflection on the above comments.
Reviewer 3 Report
Authors have reported the synthesis of SrAl2O4: Eu2+, Dy3+ powder by combustion method and show cased its application as an anti-counterfeiting raw material. Reviewer has some comments to be addressed before the manuscript is considered for publication.
The abstract should be rewritten, with some information related to the results. It is simply written as what you are doing. Provide the results related to lifetime, quantum yield etc.
The synthesis and luminescence mechanism of Eu2+, Dy3+ doped SrAl2O4 powder is not unique, it has been widely studied and reported by various methods with security related applications [New J. Chem., 2015, 39, 3380-3387]. Introduction provides mainly the synthesis related discussion on solid state reaction method and its demerits, without any significant discussion on the problems and current state of these type of luminescent materials in terms of lifetime, quantum yield and especially for anti-counterfeiting applications (which is basically the core point of the manuscript).
Line 55-56, provide suitable reference.
Line 56-57, What is the need of subject to your group activities to be stated in the manuscript. There is no need to explain what your group is doing, it well reflected from the submitted manuscript and the Funding sources with project.
Please mention, what is the optimum temperature, from Figure 1b, and discussion it is not clearly mentioned here, at which optimized temperature doping was considered. 800 or 900 C?
Please mention, what is the concentration of doping for samples used for SEM images?
Line 147, 285, is it the average particle size or scale size?
What are the particle sizes at different temperatures from SEM images, it seems they are quite irregular shaped? With Solid state methods, it’s been reported like a spherical solid ball-like fine structures connected with each other having average size of 0.5 um [New J. Chem., 2015, 39, 3380-3387], while in the present case the size seems large and irregular. Can you explain why this morphological difference between the two different methods and its impact photoluminiscence?
Line 151-152, with SEM image, can you explain the successful synthesis, if so, can you provide corresponding EDX analysis confirming the elemental composition?
Round 2
Reviewer 2 Report
The authors have successfully improved the original version of the manuscript, responding constructively to all the comments/recommendations of the reviewer. The article can be recommended for publication.
Reviewer 3 Report
Authors have revised the manuscript significantly. The manuscript could be accepted in the presrent form. Kindly check the reference sequencing in the introduction.
